# Ethephon-Induced Ethylene Enhances Protein Degradation in Source Leaves, but Its High Endogenous Level Inhibits the Development of Regenerative Organs in *Brassica napus*

**DOI:** 10.3390/plants10101993

**Published:** 2021-09-23

**Authors:** Bok-Rye Lee, Rashed Zaman, Van Hien La, Sang-Hyun Park, Tae-Hwan Kim

**Affiliations:** 1Department of Animal Science, Institute of Agricultural Science and Technology, College of Agriculture & Life Science, Chonnam National University, Gwangju 61186, Korea; turfphy@jnu.ac.kr (B.-R.L.); rashedzaman@ru.ac.bd (R.Z.); lavanhien@tuaf.edu.vn (V.H.L.); ghost1284@jnu.ac.kr (S.-H.P.); 2Asian Pear Research Institute, Chonnam National University, Gwangju 61186, Korea; 3Department of Genetic Engineering and Biotechnology, University of Rajshahi, Rajshahi 6205, Bangladesh; 4Faculty of Biotechnology and Food Technology, Thai Nguyen University of Agriculture and Forestry, Quyet Thang Commune, Thai Nguyen City 24119, Vietnam; 5Institute of Environmentally-Friendly Agriculture, Chonnam National University, Gwangju 61186, Korea

**Keywords:** amino acid transport, *Brassica napus*, ethylene, proteolysis, source-sink relationship

## Abstract

To investigate the regulatory role of ethylene in the source-sink relationship for nitrogen remobilization, short-term effects of treatment with different concentrations (0, 25, 50, and 75 ppm) of ethephon (2-chloroethylphosphonic acid, an ethylene inducing agent) for 10 days (EXP 1) and long-term effects at 20 days (Day 30) after treatment with 100 ppm for 10 days (EXP 2) on protein degradation and amino acid transport in foliar sprayed mature leaves of *Brassica napus* (cv. Mosa) were determined. In EXP 1, endogenous ethylene concentration gradually increased in response to the treated ethephon concentration, leading to the upregulation of *senescence-associated gene 12 (SAG12)* expression and downregulation of *c**hlorophyll a/b-binding protein (CAB)* expression. Further, the increase in ethylene concentration caused a reduction in protein, Rubisco, and amino acid contents in the mature leaves. However, the activity of protease and expression of amino acid transporter (*AAP6*), an amino acid transport gene, were not significantly affected or slightly suppressed between the treatments with 50 and 75 ppm. In EXP 2, the enhanced ethylene level reduced photosynthetic pigments, leading to an inhibition of flower development without any pod development. A significant increase in protease activity, confirmed using in-gel staining of protease, was also observed in the ethephon-treated mature leaves. Ethephon application enhanced the expression of four *amino acid transporter genes* (*AAP1*, *AAP2*, *AAP4*, and *AAP6*) and the phloem loading of amino acids. Significant correlations between ethylene level, induced by ethephon application, and the descriptive parameters of protein degradation and amino acid transport were revealed. These results indicated that an increase in ethylene upregulated nitrogen remobilization in the mature leaves (source), which was accompanied by an increase in proteolytic activity and amino acid transport, but had no benefit to pod (sink) development.

## 1. Introduction

Ethylene production increases drastically during several plant developmental events, such as germination, leaf and flower senescence and abscission, and fruit ripening [1,2,3,4,5]. Ethylene is involved in the regulation of plant responses to abiotic stress [6,7] and plays a critical role in the modulation of plant defense against different biotic stresses [8]. Exogenous treatments with ethylene and its precursors as well as ethylene inhibitors have demonstrated clear links between this volatile plant hormone and plant defense responses [9,10]. Among the diverse regulatory roles of ethylene, the most widely studied aspect has been its involvement in the leaf senescence process and regenerative development.

Ethylene promotes leaf senescence [3,11,12,13] and inhibits regenerative development [2,14,15]. The production of reproductive organs (sink) is closely associated with a massive, coordinated degradation of macromolecules stored in source leaves and their remobilization to sink organs. Proteolysis plays a key role in foliar N remobilization. Using ^15^N tracing, Girondé et al. [16] have estimated that the amount of N distributed to the silique was mainly derived from remobilized N from the source leaves. A large portion of remobilized N from senescent leaves is derived from the degradation of leaf proteins (mainly Rubisco) [17,18], and the amino acids released after protein degradation are transported via phloem to meet the N demand for grain filling [19,20]. Although the role of ethylene in developmentally regulated senescence has been extensively studied [3,11,12,13], its role in the regulation of N remobilization from mature leaves (source) for pod (sink) development is still not clearly established.

Several studies have shown that salicylic acid is involved in leaf senescence [13,21,22]. Similarities between the pathways of salicylic acid and senescence with regard to gene modulation have been reported [23]. Our previous studies have provided evidence of salicylic acid-mediated protein degradation and amino acid transport in mature leaves [24]. Given that ethylene inhibits the flowering process [14] and the development of regenerative organs [2,15], the reduced or removed sink strength during the regenerative stage certainly results in a corresponding change in N remobilization from mature leaves (source). However, protein degradation and amino acid transport in source leaves in response to a modification (by ethylene evolution) of the source-sink relationship have not been clearly defined.

In the present study, we hypothesized that (1) ethephon (2-chloroethylphosphonic acid) application enhances the endogenous ethylene level and (2) this modification has a significant influence on the development of regenerative organs (sink), proteolytic processes in mature leaves (source), and amino acid transport at the early pod filling stage. To test these hypotheses, morphological changes in the reproductive organs and changes in the N assimilate pool, protein profile, protease activity, amino acid transporter gene expression, and phloem loading of amino acid in mature leaves (source) were compared between the ethephon-treated plants and the controls through serial experiments, short-term effects after treatment with four different concentrations of ethephon (0, 25, 50, and 75 ppm) for 10 days, and long-term effects at 20 days (day 30) after 10 days of treatment with 100 ppm.

## 2. Results

### 2.1. Short-Term Effects of Ethephon Treatment with Different Concentrations

#### 2.1.1. Plant Morphology, *CAB*, and *SAG12* Gene Expression

Different concentrations (0, 25, 50, and 75 ppm) of ethephon were sprayed to foliar leaves of *Brassica napus* for 10 days. At all concentrations of ethephon, foliar ethephon spraying significantly reduced stem elongation and flowering. The height of the plant and the number of branches and flowers gradually decreased with the increasing concentration of ethephon (Figure 1A). There were few morphological changes at concentrations up to 25 ppm of ethephon, whereas a distinct inhibition of branch development and flowering, along with browning, wilting, and even drying leaves, was observed at 75 ppm. Foliar ethephon spraying induced endogenous ethylene in mature leaves. Endogenous ethylene concentration was increased in an ethephon concentration-dependent manner, with a 1.5- and 1.7-fold increase, respectively, in ethylene level in 50 ppm or 75 ppm ethephon-treated mature leaves compared with that in the control at day 10 (Figure 1B). *Chlorophyll a/b-binding protein* (*CAB*) gene expression was highly suppressed in 50 and 75 ppm ethephon-treated leaves (Figure 1C). In contrast, *senescence-associated gene 12* (*SAG12*) expression enhanced progressively in response to the applied ethephon concentration (Figure 1D).

#### 2.1.2. Protein Profiles, Protease Activity, and Active Staining

Protein content in control leaves significantly decreased after 10 days. At 10 days after ethephon treatment, a progressive decrease in protein content was observed in response to the ethephon-treated concentration (Figure 2B). However, the decrease in protein was not significant at concentrations up to 25 ppm, whereas it was significant in 50 ppm or 75 ppm ethephon-treated mature leaves, in agreement with SDS-PAGE protein profiles (Figure 2A,B). The higher concentration (over 50 ppm) of ethephon treatment markedly reduced the expression of Rubisco large subunit and small subunit (Figure 2A).

Protease activity was gradually increased 1.7-fold, 2.7-fold, or 2.4-fold in 25, 50, or 75 ppm ethephon-treated mature leaves, respectively, compared with the control plants mature leaves at day 10 (Figure 2C). Three isozymes of protease were expressed in mature leaves. The protease 1 expression enhanced in a treated-ethephon-dependent manner, whereas the proteases 2 and 3 were expressed only in 50 and 75 ppm ethephon-treated mature leaves (Figure 2D).

#### 2.1.3. Amino Acid Content and Amino Acid Transport

Amino acid content in mature leaves in control leaves significantly decreased after 10 days. At 10 days after ethephon treatment, the ethephon-induced decrease in amino acid content was significant only in 50 ppm (−17.8%) and 75 ppm (−30.2% compared with control) (Figure 2E). The expression of an amino acid transporter (amino acid permease 6, *AAP6*) was progressively enhanced at concentrations up to 50 ppm and was slightly suppressed in the 75 ppm ethephon-treated mature leaves, but this change was not significant (Figure 2F).

### 2.2. Long-Term Effects of Ethephon Treatment

Considering that a decrease in protease activity and protein content in mature leaves was no more activated in 75 ppm ethephon treatment compared with those in 50 ppm ethephon treatment, long-term effects (e.g., at 30 days) of ethylene, which were induced by 100 ppm ethephon for 10 days, on protein degradation in source leaves and sink development (pod formation) were assessed especially in relation to morphological changes.

#### 2.2.1. Plant Morphology and Biomass

At 10 days, the stem elongation of the apical part of 100 ppm ethephon-treated plants was remarkably suppressed when compared with that of the control. Normal flower development and newly formed leaves were observed on the elongated branches of the control plants, whereas immature floral buds and newly formed leaves were crowded on the upper part of the stems of the ethephon-treated plants (Figure 3B; Table 1). After the additional 20 days (day 30), the stem elongation and branch development were still markedly lower in the ethephon-treated plants than in the control (Figure 3C; Table 1). Normal pods and some newly formed leaves were developed on the newly formed large and long branches in the control plant, whereas large numbers of newly formed leaves gathered on a few small branches, without any pod development, in the ethephon-treated plants (Figure 3C; Table 1).

Ethephon treatment significantly affected the biomass of most plant tissues (Table 1). The biomass of fallen leaves was significantly increased, especially at day 30. The biomass of mature leaves decreased by 44.2% in the ethephon-treated plants when compared with that of the control at day 30. The biomass of newly formed leaves in the ethephon-treated plants at day 30 was significantly greater (+76.4%) than that in the control. The biomass of flowers and immature floral buds at day 10 was significantly lower (−80%) in the ethephon-treated plants than in the control. Pod development was observed only in the control plant (8.3 g plant^−1^) at day 30. Ethephon treatment significantly decreased root biomass by 37.1% and 45.3% at day 10 and day 30, respectively.

#### 2.2.2. Ethylene, Total Chlorophyll, and Carotenoid Content

Endogenous ethylene concentration was 88.9% and 46.6% higher in the ethephon-treated plants than that of the control at day 10 and day 30, respectively (Figure 3D). The ethephon treatment resulted in a reduction in photosynthetic pigments. The total chlorophyll content in the ethephon-treated mature leaves decreased by 14.4% and 37.9% at day 10 and day 30, respectively, compared with that of the control (Figure 3E). Similarly, the carotenoid content in the mature leaves also decreased by 17.9% and 37.7% at day 10 and day 30, respectively, compared with that of the control (Figure 1F).

#### 2.2.3. Soluble Protein and Amino Acids Content

Ethephon treatment decreased soluble protein content in the mature leaves by 46.9% and 44.5% at day 10 and day 30, respectively (Table 2). In contrast, in the newly formed leaves, the soluble protein content increased by 18.4% and 68.01% compared with that of the control at day 10 and day 30, respectively (Table 2). The soluble protein content in pods, which was produced only in the control plant, was 5.58 mg g^−1^ FW at day 30. Ethephon treatment decreased amino acid content in the mature leaves by 21.9% and 14.6% at day 10 and day 30, respectively (Table 2). In the newly formed leaves, the amino acids content was significantly higher for the ethephon-treated plants than for the control plants. The amino acids content of pods in the control plant was 2.67 mg g^−1^ FW at day 30. The amino acid content in the phloem exudates in the ethephon-treated plant was 55.8% and 65.5% higher than that of the control at day 10 and day 30, respectively (Table 2).

#### 2.2.4. SDS-PAGE Protein Profile, Protease Activity, and Amino Acid Transporters

The SDS-PAGE protein profiles (Figure 4A) reveled that ethephon treatment suppressed protein expression, especially bands in 50 kDa (Rubisco large sub-unit) and 15 kDa (Rubisco small sub-unit). The protease activity in the ethephon-treated mature leaves was 70.5% and 58.5% higher than that of the control at day 10 and day 30, respectively (Figure 4B). Ethephon treatment significantly enhanced the expression of four amino acid transporter genes in the mature leaves. At day 10, the expression of amino acid permease was significantly enhanced by 3.7-, 3.0-, 4.6-, and 4.6-fold for *AAP1*, *AAP2*, *AAP4*, and *AAP6*, respectively (Figure 4C).

#### 2.2.5. Heatmap Responses of Pearson’s Correlation Coefficient (r) among the Metabolites or Gene Expression

To further examine the functional implications and correlations of the identified metabolites or gene expression levels as affected by ethephon, we created heatmap responses of Pearson’s correlation coefficient (r) among protein, amino acids, protease, and amino acid transporters. 

A positive correlation was observed between amino acid transporters and amino acid content in newly formed leaves, whereas a negative correlation was observed between protease activity and protein content in mature leaves (Figure 5A). A comparative analysis of the factors related to ethylene content (presented by green boxes) suggested that the correlation of ethylene was revealed to be positive with protein and amino acid content in newly formed leaves, and expression of amino acid transporters *AAP1, AAP2, AAP4,* and *AAP6*, and negative with the protein and amino acid contents in mature leaves (Figure 5B).

## 3. Discussion

A source-sink relationship is generally defined as nutrient mobilization from the mature leaves (source) to the nutrient-receiving organs (sink). The remobilization of N assimilates from source leaves is a crucial determinant for the production of the reproductive organs (sink), especially in *B. napus*, which has low nitrogen remobilization capacity, with less than 50% of nitrogen stored in source leaves used for seed filling [25,26]. Ethylene is the active signal compound involved in the senescence process of source leaves and in the sink development [2,11,12,14]. In this context, the present study emphasized the regulatory role of ethylene in protein degradation and amino acid transport in mature leaves (source) in relation to pod (sink) development in response to ethephon treatment at the early regenerative stage of *B. napus.*

### 3.1. Ethephon-Induced Ethylene Effects on Morphological Change

Ethephon (2-chloroethyl phosphonic acid) induces the release of ethylene and accelerates endogenous ethylene production. Thus, it is widely used as a chemical replacement for ethylene treatment in many physiological studies [27]. Ethephon-induced ethylene production, accompanied by an increase in enzyme 1-aminocyclopropane-1-carboxylase synthase (ACS) activity, has been reported [28,29]. As expected, in the present study, foliar spraying with 25, 50, 75, and 100 ppm ethephon resulted in an increase in the endogenous level of ethylene (Figure 1B and Figure 3D). Morphological changes in ethephon-treated plants were characterized by decreased plant length, reduced root biomass, abscission of lower leaves, and formation of new leaves around the apex (Figure 1A and Figure 3B,C; Table 1). Based on visual observations, the decrease in plant height was due to an inhibition of internode elongation, which is responsible for the production of the branches that form flowers and pods. Previous studies in many plant species revealed that ethylene inhibited flowering and reproductive bud development as a result of repression of the floral meristem-identity gene LEAFY via ethylene-induced DELLA accumulation [30], showing the flower bud abortion and abscission of flower petals [1,2]. However, less or no reduction of branch, flowers, and internode elongation was observed in 25 ppm ethephon-treated plants (Figure 1A). These results indicate that ethephon-induced ethylene governs the development of leaves, branches, and flowers, as well as stem elongation in an applied concentration-dependent manner. Several reports have shown that external ethephon application has an influence on the endogenous ethylene evolution, which is involved in regulating plant growth and development depending on its concentration, timing of application, and the plant species [31,32].

### 3.2. Ethephon-Induced Ethylene-Mediated Leaf Senescence

The present study confirmed that ethephon-induced ethylene is a promoter of leaf senescence, as shown by the increase in the number of senescent and fallen leaves (Table 1), the decrease in photosynthetic pigments (Figure 3), and a gradual enhancement of *senescence-associated gene* (*SAG12*) expression (Figure 1C), which has been widely observed during leaf senescence of *B. napus* [33,34,35]. The most common visible symptom of leaf senescence is the yellowing caused by the chlorophyll degradation and its impaired biosynthesis. In the present study, the depression of *chlorophyll a/b-binding protein* (*CAB*) gene was remarked in 50 and 75 ppm ethephon-treated mature leaves (Figure 1C), leading to a decrease in total chlorophyll content (−37.9% compared with the control) at day 30 (Figure 3E). Numerous studies have shown that exogenous application of ethylene or ethylene releaser accelerates the leaf senescence process [3,12,13]. Ethylene has been shown to be involved in the organized cell dismantling process, which includes nucleic acid reduction, protein degradation [36,37], lipid degradation, peroxidation [38,39], and the activation of nutrients recycling from senescing leaves to the other organs [17,19]. The premature senescence of green leaves in soybean plants, induced by ethylene, directly resulted in a decrease in source strength and a subsequent decrease in pod development [6]. It has been documented that leaf senescence leads to a massive coordinated degradation of macromolecules and nutrient remobilization from source leaves for the ultimate phase of pod development [37,40]. In many plant species, the source activity drives the sink metabolism, and vice versa, which in turn are related to C and N metabolism [24,41]. Proteolysis in the senescent leaf is the most important process for foliar nitrogen remobilization, as it modifies the source-sink relationship with an increase in nitrogen remobilization in source leaves [42]. Previous studies have shown that a large portion of N remobilization from senescent leaves is derived from the degradation of leaf proteins (mainly Rubisco) [17,18]; amino acids released after protein degradation are transported, via phloem, to meet the N demand for grain filling [19,20]. Leaf senescence is a genetically programmed developmental process, which shifts sink strength from younger leaves to developing pods [37]. Thus, during normal development in the early regenerative stage, protein degradation in mature or senescent leaves (source), and the transport of the released amino acids, are coordinated with the demand of assimilates for pod development and seed filling (sink), in order to maintain the balance between supply and demand.

### 3.3. Ethephon-Induced Ethylene Effects on Protein Degradation and Amino Acid Transport

In these contexts, we hypothesized that the total inhibition of pod development (i.e., the decreased or removed sink strength), caused by the increased ethylene level in ethephon-treated plants, could delay the senescence process. This may be liable for the reduction in protein degradation in mature source leaves as well as the transport of amino acids. However, the present study indicated that the enhanced ethylene level, caused by the foliar ethephon spray, strongly activated the breakdown of proteins, especially Rubisco (Figure 2A and Figure 4A), which is presumably a major compound subjected to proteolysis, and serves as the most N remobilized protein during leaf senescence [2,18]. The enhanced proteolytic activity in ethephon-treated mature leaves was confirmed by the increased activity of protease (Figure 2C and Figure 4B), in agreement with the in-gel staining of proteases (Figure 2D). However, protein degradation was not observed in a lower concentration of ethephon-treated plants (25 ppm), even though protease activity was slightly increased (Figure 2C). Girondé et al. [16] reported that leaf senescence, induced by nitrogen deficiency, enhanced protein degradation, accompanied by a large increase in the activity of cysteine proteases and in the transcript level of *SAG12* encoding senescence-specific cysteine-protease, which is closely associated with the remobilization of leaf proteins [43]. The increase in *SAG12* at the transcriptomic and proteomic levels has been widely observed during leaf senescence of *B. napus* [33,44]. As previously reported, ethephon treatment-induced protein degradation was closely associated with the upregulation of SAG12 gene expression, as ethephon concentration was increased, a parallel with the enhancement of cysteine-protease activity (Figure 2C and Figure 4B) and degradation of Rubisco large or small subunit (Figure 2A and Figure 4A).

Our previous study showed that an increase in the salicylic acid level, by exogenous application, was involved in leaf senescence and enhanced proteolytic activity [24]. In accordance with these observations, the present study confirmed that the high concentration of ethylene also accelerates the senescence process of mature leaves, as shown by the higher degradation of leaf proteins and enhanced activity of proteases. Amino acids released after protein degradation are transported, via phloem, to meet the N demand for grain filling [19,20]. To elucidate the role of ethylene in regulating amino acid transport, we analyzed the expression of amino acid transporter genes and amino acid content in phloem exudates. The resulting data showed that the increased level of ethylene, induced by ethephon, up-regulated the expression of four amino acid permeases, *AAP1*, *AAP2*, *AAP4,* and *AAP6* (Figure 2F and Figure 4C). Especially, the expression level was remarkably higher in 50 ppm ethephon-treated plants than other ethephon treatments, which was consistent with protease activity (Figure 2F). In *Arabidopsis*, *AtAAP1* and *AtAAP2* were strongly expressed in siliques during fruit ripening [45]. In *B. napus*, *BnAAP1* was strongly expressed in the flowers and leaves, and *BnAAP2* in the stem at low N supply [46]. The expressions of *BnAAP1, BnAAP2*, and *BnAAP6* were upregulated in the mature leaves as a result of an increase in salicylic acid [24]. Tilsner et al. [46] suggested that *BnAAP6,* which is mostly expressed in leaves and stems, is responsible for the phloem loading of amino acids. In the present study, the content of amino acid in phloem exudates was also significantly higher in mature leaves of the ethephon-treated plants (Table 2). Moreover, linear correlations between the ethylene concentration (as affected by the foliar application of ethephon) and protease activity, the expression of amino acid transporter genes, and the amino acid content in phloem exudates were also significant (Figure 5).

### 3.4. High Level of Ethylene Supressed Regenerative Organ Development and Unintended Source-Sink Relationship

Given that, at 20 days after 100 ppm ethephon-treatment for 10 days (day 30), the newly formed leaves gathered around the apex (e.g., dwarfism with nearly half of plant height) without any development of pod-holding branches (Figure 3C), the results presented above clearly indicated that N remobilization from the mature source leaves continued although the main sink strength for the pod development was nearly null. Based on the present data, direct evidence of high ethylene-induced dwarfism is not clear, but could be explained by the interaction with other hormones’ regulation of this process. For instance, ethylene inhibits auxin transport in the veinal tissues [47,48], and reduction of auxin in the shoot apex depresses the growth of stem internodes, which is a strong sink, leading to a stimulation of apical bud development by supplying sugars necessary for their development [49]. Ethylene can trigger the senescence in an interaction with ABA [50], which eventually regulates the starch degradation in source leaves and phloem sucrose loading [35]. Therefore, the dwarfism and the gathering of newly formed leaves at a high endogenous ethylene level might be an unintended consequence of source-sink relationships and result from an overflow of C and N assimilates to axillary buds that cannot be utilized by the main shoot to develop regenerative organs as in the control plants (Figure 3B,C). It has been reported that the higher amount of amino acids in phloem is closely associated with outgrowth of axially bud when growing shoot tip was removed [51,52]. In this context, we estimated the amino acid content in phloem exudates in order to assess N remobilization into the potent sink organs (i.e., newly produced leaves and pods). As expected, ethephon treatment with a high concentration (100 ppm) resulted in an enhancement in phloem amino acid loading by 55.8% and 65.5% at day 10 and day 30, respectively (Table 2), leading to a significant increase in amino acid content in newly formed leaves compared with that of the control (Table 2). It thus suggested that the newly formed leaves in the ethephon-treated plants acted as a potent sink for the amino acids released from the protein degradation in the source leaves, as an unintended source-sink relation.

## 4. Materials and Methods

### 4.1. Plant Culture and Experimental Designs

The seedlings of *Brassica napus* (cv. Mosa) were grown in 2 L pots with daily feeding with complete nutrient solution [53]. EXP 1 and 2 were conducted for the short- or long-term effects of ethephon, respectively, in mature leaves of *B. napus*. For the experiments, plants were selected based on morphological similarities and divided at the beginning of the bolting stage. For EXP1, plants were foliar sprayed with 50 mL of different concentrations (0, 25, 50, and 75 ppm) of ethephon (as an ethylene inducer) or the same volume of water (control) for 10 days. For EXP2, one group was foliar sprayed with 50 mL of 100 ppm ethephon twice per day for 10 days, whereas the other group (control) was sprayed with the same volume of water for 10 days. After 10 days (day 10), half of the control and ethephon-treated plants were sampled; the rest were kept for an additional 20 days (day 30). After harvest, the leaves were separated in order of ontogenic appearance and designated a leaf rank number (i.e., rank 1 for the oldest leaf). In this study, the mature leaves that were ranked 4–12 were considered source leaves and the newly developed tissues (newly formed leaves and pods) after ethephon application were considered sink tissues. Sampled tissues were frozen immediately in liquid nitrogen and stored in a deep freezer (−80 °C) until further analysis.

### 4.2. Determination of Ethylene

The evolution of ethylene was measured using a modification of the methods described by Iqbal et al. [54]. Small tissue pieces (approximately 2 g) were placed into 10 mL tubes containing moist paper to minimize evaporation from the tissue, and the tubes were rapidly sealed using rubber caps. The sealed vials were then incubated in a dark growth chamber at 25 °C for 24 h. One hundred microliter of gas was withdrawn with a hypodermic syringe, and the ethylene concentration was analyzed on a gas chromatograph (Agilent 7890A, Agilent technologiesInc., DE, USA) equipped with a frame ionization detector. The columns were HP-5, 30 m, 0.32 mm, and 0.25 μm. The temperatures of the column oven, injector, and detector were 100, 200, and 250 °C, respectively. The flow rate of the carrier gas (helium, 99%) was 25 mL min^−1^. Ethylene was identified based on the retention time and quantified by comparison with picks from the standard ethylene concentration.

### 4.3. Determination of Photosynthetic Pigments

Leaf tissues (approximately 200 mg) were placed in 10 mL dimethyl sulfoxide for each sample and kept under dark conditions at 25 °C for 2 days and then incubated at 65 °C for 30 min. After incubation, absorbance was checked at 480, 510, 663, and 645 nm. Total chlorophyll and carotenoid contents were calculated using the following equations:Total chlorophyll (mg L^−1^) = 20.2 *A*_645_ + 8.02 *A*_663_
Carotenoid (mg L^−1^) = 7.6 *A*_480_ + 1.49 *A*_510_

### 4.4. Collection of Phloem Exudates and Determination of Amino Acids and Protein

The EDTA-facilitated method was used for the collection of phloem exudates, as described by Lee et al. [55]. The resultant collection was stored in a deep freezer (−80 °C) until further analysis. Amino acid concentrations were measured using the ninhydrin colorimetric method [56]. Briefly, fresh ground samples (approximately 100 mg) were extracted with 100 mM KPO_4_^−^ (pH 7.0) and centrifuged at 12,000× *g* for 10 min at 4 °C. The supernatant was then mixed with ninhydrin solution and boiled for 10 min, after which it was cooled on ice. Fifty percent of ethanol was added and mixed; then, absorbance of the reaction mixture was recorded at 570 nm. Protein concentration was determined using the Bradford reagent (Sigma B6916, St. Louis, MI, USA), with bovine serum albumin as the standard.

### 4.5. Protein Profiles by SDS-PAGE

To study protein profiles, SDS-PAGE was performed in a mini vertical electrophoresis system (Bio-Rad, Mini-PROTEAN, CA, USA). Equal quantities of protein (25 µg) from each sample were loaded into 12.5% gels, and Precision Plus Protein Dual Color Standards (Bio-Rad) were incorporated into the gel to determine the molecular weight of the bands.

### 4.6. Total Protease Activity and In-Gel Staining of Protease

Total proteolytic activity and visualization of protease were determined using the method described by Beyene et al. [57]. Fresh grind sample was extracted by 50 mM Tris-HCl (pH 9.0) 12,000× *g* for 10 min at 4 °C. The supernatant was then mixed with 2% azocasein and incubated at 37 °C for 30 min. After incubation, the reaction was stopped by adding an equal volume of ice-cold 10% trichloroacetic acid, incubated at 4 °C for 10 min, and then centrifugated 12,000× *g* for 5 min. After centrifugation, the supernatant was mixed with an equal volume of 1 M NaOH for color development and the absorbance was recorded at 660 nm. One unit of enzyme activity was equal to the conversion of 1 μmol of substrate per min. For visualization of proteases, 50 µg of protein was separated on 12.5% SDS-PAGE containing 0.1% gelatin. After electrophoresis, gel was re-natured in 2.5% Triton X-100; rinsed; and developed for overnight in staining solution containing 0.125% coomassie brilliant blue R-250, 10% acetic acid, and 25% methanol.

### 4.7. RNA Extraction and Quantitative PCR

Total RNA was isolated from 100 mg of leaf tissue using a Total RNA Isolation System (Promega, Madison, WI, USA). The first-strand cDNAs were synthesized using the GoScript Reverse Transcription System (Promega). The gene expression level was quantified on a light cycler real-time PCR detection system (Bio-Rad) with SYBR Premix Ex TaqTM (TaKaRa, Kyoto, Japan). The sequences of gene-specific primers used for the qRT-PCR are presented in Appendix A. The qRT-PCR reactions were performed in duplicates for the three biological replications of each treatment. The relative expression level of target genes was calculated from threshold values (Ct), using actin as the internal control.

### 4.8. Statistical Analysis

A completely randomized design was used with three replicates for each treatment. Duncan multiple range test for EXP 1 and Student’s *t* test for EXP 2 were used to compare the means of the three replicates. Statistical analysis of all measurements was carried out using the software SAS 9.1.3 (SAS Institute Inc., Cary, NC, USA).

## 5. Conclusions

The present study suggested that ethephon-induced ethylene accelerated the senescence process of mature leaves with enhanced proteolytic activity and amino acid transport, and that over-optimal ethylene inhibited the development of regenerative organs (e.g., floral buds and pods), leading to an alternative development of new leaves at the apexes, as an unintended consequence of the source-sink relationship for the amino acids released from source leaves. Underlining the unintended source-sink relationships in the high endogenous ethylene level, a critical question, “how does ethylene regulate the source-sink sugars status in relation to intrinsic and environmental factors besides the interaction with other hormones?” needs to be answered in future works.

## Figures and Tables

**Figure 1 plants-10-01993-f001:**
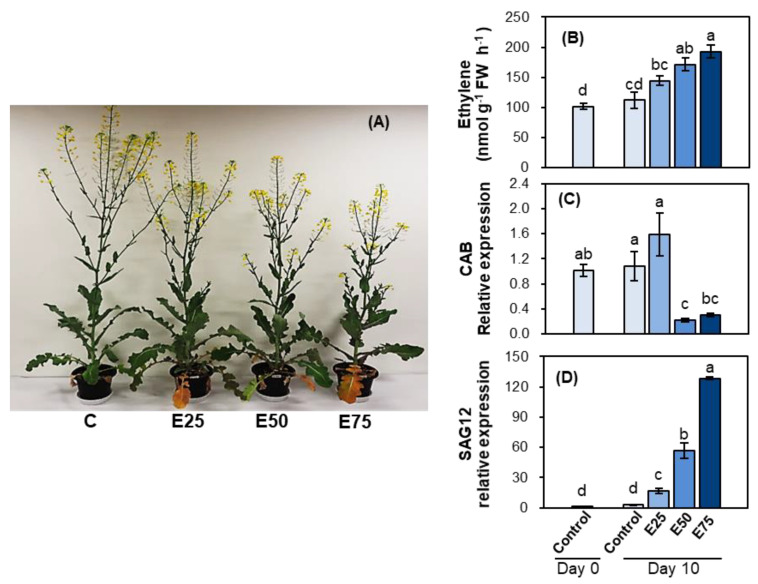
Short–term effects of different concentrations (0, 25, 50, or 75 ppm) of ethephon application on plant morphology, ethylene content and leaf senescence. (**A**) Plant morphology, (**B**) ethylene content, (**C**) *c**hlorophyll a/b*–*binding protein* (*CAB*), and (**D**) *senescence-associated gene 12* (*SAG12*) gene expressions in mature leaves of the control and ethephon-treated plants. Ethephon (0, 25, 50, or 75 ppm) was applied to foliar leaves of *Brassica napus* for 10 days. E25, E50, and E75 indicate foliar applied ethephon concentrations of 25, 50, and 75 ppm, respectively. Data are presented as mean ± SE (*n* = 3). The qRT–PCR was performed in duplicate for each of the three independent biological samples. Different letters on columns indicate significant different (*p* < 0.05) between treatments.

**Figure 2 plants-10-01993-f002:**
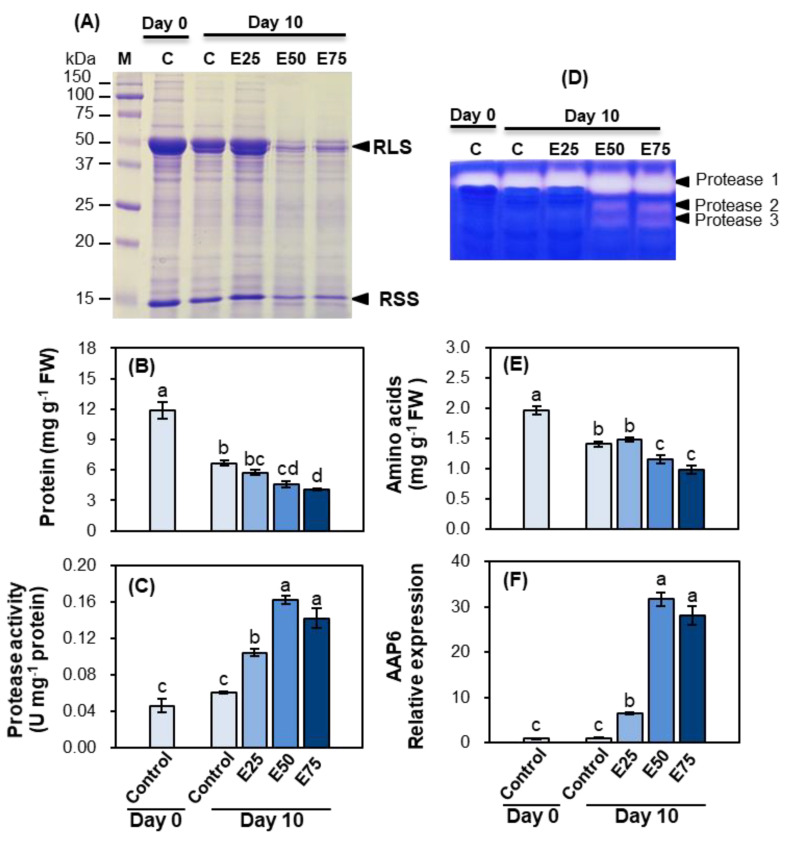
Short–term effects of different concentrations (0, 25, 50, or 75 ppm) of ethephon application on protein degradation and amino acid transport. (**A**) SDS–PAGE protein profiles, (**B**) protein content, (**C**) protease activity, (**D**) protease active staining, (**E**) amino acid content, and (**F**) amino acid transporter (amino acid permease, *AAP6*) gene expression in mature leaves of the control and ethephon–treated plants. Ethephon (0, 25, 50, or 75 ppm) was applied to foliar leaves of *Brassica napus* for 10 days. E25, E50, and E75 indicate foliar applied ethephon concentrations of 25, 50, and 75 ppm, respectively. Data are presented as mean ± SE (n = 3). The qRT–PCR was performed in duplicate for each of the three independent biological samples. Different letters on columns indicate significant different (*p* < 0.05) between treatments.

**Figure 3 plants-10-01993-f003:**
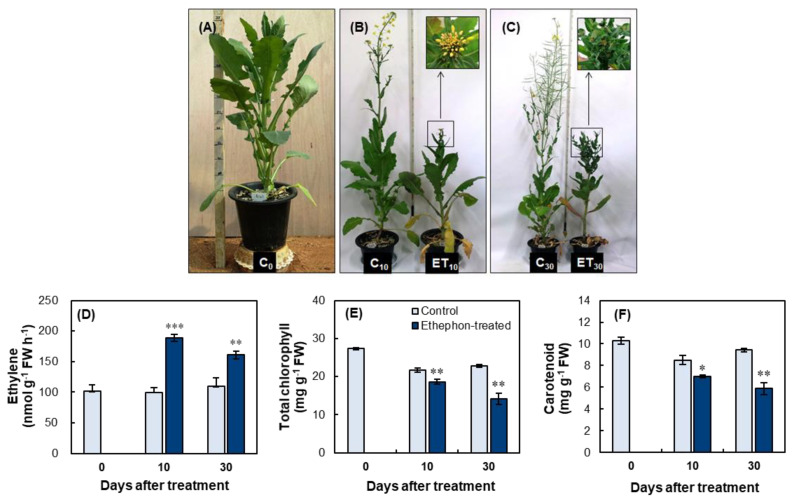
Long–term effects of 100 ppm ethephon application on plant morphology, ethylene content, and leaf senescence. (**A**–**C**) Plant morphology changes before the ethephon application (day 0, (**A**)), at the end of ethephon application for 10 days (day 10, (**B**)), and an additional 20 days after the treatment (day 30, (**C**)). (**D**) Ethylene, (**E**) total chlorophyll, and (**F**) carotenoid contents in mature leaves of the control and ethephon-treated plants. C_0_, C_10_, and C_30_ indicate the control plants at day 0, day 10, and day 30, respectively, whereas ET_10_ and ET_30_ indicate the ethephon-treated plants at day 10 and day 30, respectively. Data are presented as mean ± SE (*n* = 3). Significant differences between treatments are represented by * *p* < 0.05; ** *p* < 0.01; *** *p* < 0.001.

**Figure 4 plants-10-01993-f004:**
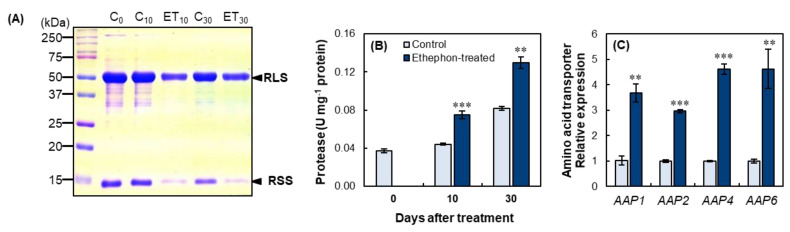
Long–term effects of 100 ppm ethephon application on protein degradation and amino acid transport. (**A**) SDS–PAGE protein profiles, (**B**) protease activity, and (**C**) amino acid transporters gene expression in mature leaves of the control and ethephon−treated plants before the ethephon application (day 0), at the end of ethephon application for 10 days (day 10), and an additional 20 days after the treatment (day 30). Each lane was loaded with 25 µg of soluble protein. Molecular mass makers (kDa) are listed on the left or right side of the gel. C_0_, C_10_, and C_30_ indicate the control plants at day 0, day 10, and day 30, respectively, whereas ET_10_ and ET_30_ indicate the ethephon-treated plants at day 10 and day 30, respectively. Data are presented as mean ± SE (*n* = 3). The qRT–PCR was performed in duplicate for each of the three independent biological samples. Significant differences between treatments are represented by ** *p* < 0.01; *** *p* < 0.001.

**Figure 5 plants-10-01993-f005:**
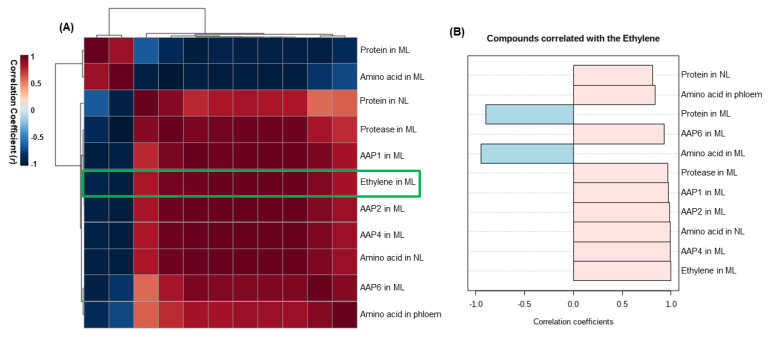
Heatmap of the overall correlations among the variables measured at day 10. (**A**) Heatmap responses of Pearson’s correlation coefficient (*r*) for the identified metabolites or gene expression levels measured in 100 ppm ethephon−treated or control plants. Red indicates positive effect, whereas blue indicates negative effect. (**B**) The factors correlated with ethylene. ML, mature leaves; NL, newly formed leaves.

**Table 1 plants-10-01993-t001:** Plant height, stem width, and biomass of different tissues for the control and ethephon-treated plants before the ethephon application (day 0), at the end of ethephon application for 10 days (day 10), and an additional 20 days after the treatment (day 30). Data are presented as mean ± SE (*n* = 3).

	Days after Treatment				
	Day 0	Day 10		Day 30	
	Control	Control	Ethephon-Treated	Control	Ethephon-Treated
Plant height (cm)	25.0 ± 2.9	133.7 ± 2.4	72.0 ± 7.0 **	171.3 ± 4.4	94.5 ± 2.4 ***
Stem diameter (cm)	1.13 ± 0.03	1.50 ± 0.06	1.83 ± 0.15 *	1.80 ± 0.06	1.70 ± 0.10
Biomass (g plant^−1^)					
Fallen leaves	0.31 ± 0.10	0.34 ± 0.10	1.10 ± 0.17 *	2.62 ± 0.10	3.82 ± 0.29 *
Mature leaves	6.58 ± 0.62	7.71 ± 0.63	5.94 ± 1.07	4.71 ± 0.09	2.63 ± 0.10 ***
Newly formed leaves	1.82 ± 0.11	3.14 ± 0.10	3.14 ± 0.57	3.69 ± 0.09	6.51 ± 0.13 ***
Flowers or floral buds	–	0.50 ± 0.04	0.11 ± 0.01 ***	–	–
Pods	–	–	–	8.30 ± 0.52	–
Roots	2.45 ± 0.53	5.44 ± 0.39	3.40 ± 0.22 *	12.23 ± 0.47	6.69 ± 0.49 ***

Significant differences between treatments are represented by * *p* < 0.05; ** *p* < 0.01; *** *p* < 0.001. “–“means not measured parameter.

**Table 2 plants-10-01993-t002:** Soluble protein and amino acid content in mature leaves, newly formed leaves, pods, and phloem exudates for the control and ethephon−treated plants before the ethephon application (day 0), at the end of ethephon application for 10 days (day 10), and an additional 20 days after the treatment (day 30). Data are presented as mean ± SE (*n* = 3).

	Days after Treatment				
	Day 0	Day 10		Day 30	
	Control	Control	Ethephon-Treated	Control	Ethephon-Treated
Soluble protein					
Mature leaves (mg g^−1^ FW)	12.47 ± 0.51	9.50 ± 0.85	5.04 ± 0.57 *	7.14 ± 0.71	3.96 ± 0.09 **
Newly formed leaves (mg g^−1^ FW)	10.75 ± 0.69	8.80 ± 0.07	10.42 ± 0.51 *	4.72 ± 0.36	7.93 ± 0.19 ***
Pods (mg g^−1^ FW)	–	–	–	5.58 ± 0.17	–
Amino acids					
Mature leaves (mg g^−1^ FW)	1.26 ± 0.01	1.14 ± 0.01	0.89 ± 0.04 **	1.03 ± 0.03	0.88 ± 0.01 ***
Newly formed leaves (mg g^−1^ FW)	2.48 ± 0.12	2.23 ± 0.01	2.74± 0.02 ***	1.77 ± 0.02	2.28 ± 0.11 *
Pods (mg g^−1^ FW)	–	–	–	2.67 ± 0.10	–
Phloem exudates (µg mL^−1^ h^−1^)	45.07 ± 0.85	31.49 ± 1.52	49.06 ± 2.08 *	26.90 ± 0.39	44.52 ± 0.57 ***

Significant differences between treatments are represented by * *p* < 0.05; ** *p* < 0.01; *** *p* < 0.001. “–“means not measured parameter.

## Data Availability

The data presented in this study are available on request from the corresponding author.

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
