# Peer review of "Ethephon-Induced Ethylene Enhances Protein Degradation in Source Leaves, but Its High Endogenous Level Inhibits the Development of Regenerative Organs in Brassica napus"

_plants, 2021, doi:10.3390/plants10101993_

Round 1

Reviewer 1 Report

This article should be accepted after minor revision in several points.

>Title: should consider revise to complex and easy to understand

>Abstract:
- Should add plant name (Brassica napus)
- (Experiment 1) and (Experiment 2) are difficult to understand. Exp 1 and Exp 2 should move to the beginning of sentence.

>Results
-2.1.1: should add 1-2 sentence in brief of method.
-Fig.3: please add "total" in front of chlorophyll
-Table 1
         * Does "stem width" means "stem diameter"?
         * Please change "dead leaf" to "senescence leaf".
         * Please add what "-" mean ?

>Materials and methods
-4.1.1: Line 405-408 is same as your brief report published in Plants. Please rewrite or paraphrase. 

Author Response

Authors’ responses to reviewer’s comments

The authors deeply appreciate for Reviewers’ critical reading. The manuscript now has been revised thoroughly according to the comments and fully addressed to the questions.

Reviewer 1

This article should be accepted after minor revision in several points.
Title: should consider revise to complex and easy to understand
Response) As suggested, the title of manuscript has been changes to “Ethephon-induced ethylene enhances protein degradation in source leaves, but its high endogenous level inhibits development of regenerative organs in Brassica napus” (see the Title in the revised manuscript).

Abstract:
- Should add plant name (Brassica napus)

Response) Plant name was added in abstract in the revised manuscript (see line 25 in the revised manuscript).

- (Experiment 1) and (Experiment 2) are difficult to understand. Exp 1 and Exp 2 should move to the beginning of sentence.
Response) As suggested, “(Experiment 1) and (Experiment 2)” were replaced by “EXP1 and EXP 2” (see lines 23-25 and 31 in the revised manuscript).

Results
-2.1.1: should add 1-2 sentence in brief of method.

Response) The sentence about the method of ethephon treatment was added at the beginning of the results (see lines 91-92 in the revised manuscript).

-Fig.3: please add "total" in front of chlorophyll
Response) “chlorophyll” was changed to “total chlorophyll” (see lines 178, 186, 189, and 300 in the revised manuscript).

-Table 1
* Does "stem width" means "stem diameter"?

Response) "stem width" changed to "stem diameter" (see Table 1 in the revised manuscript).

* Please change "dead leaf" to "senescence leaf".
Response) “dead leaf” means fallen leaves. “dead leaf” was revised to “fallen leaves” in the revised manuscript (see line 167 and Table 1 in the revised manuscript).

* Please add what "-" mean ?
Response) “-“ means not measured parameter was presented the blank indicated as “-“. For example, flower or floral buds was not appeared at day 0.

Materials and methods
-4.1.1: Line 405-408 is same as your brief report published in Plants. Please rewrite or paraphrase. 

Response) The sentence has been revised to “The seedlings of Brassica napus (cv. Mosa) were grown in 2 L pots with daily feeding with complete nutrient solution [53].” in the revised manuscript (see lines 409-411 in the revised manuscript).

Reviewer 2 Report

The paper intend to study the regulatory role of ethylene in the source–sink relationship for nitrogen remobilization, short term effects of treatment with different concentrations of an ethylene inducing agent ethephon to check protein degradation and amino acid transport in foliar sprayed mature leaves of Brassica napus. Their key results from this study demonstrated that an increase in ethylene upregulated nitrogen remobilization in the mature leaves (source), which was accompanied by an increase in proteolytic activity and amino acid transport, but had no benefit to pod (sink) development. In general, the paper was well-written and results is quite significant, will be suitable published in the journal with the miner revision.

Minor comments:

  1. Literature review should be updated. There are almost no latest papers cited from 2018-2021 (only one from both 2019 and 2020) in this paper. Author cited the latest paper in this point of view in the introduction part.
  2. All the publication year was blacked in reference list except literature 25 and 27. Reference 26 was missing.

Author Response

作者对审稿人意见的回应

作者对审稿人的批判性阅读深表感谢。手稿现在已经根据评论进行了彻底的修改,并充分解决了问题。

审稿人 2

The paper intend to study the regulatory role of ethylene in the source–sink relationship for nitrogen remobilization, short term effects of treatment with different concentrations of an ethylene inducing agent ethephon to check protein degradation and amino acid transport in foliar sprayed mature leaves of Brassica napus. Their key results from this study demonstrated that an increase in ethylene upregulated nitrogen remobilization in the mature leaves (source), which was accompanied by an increase in proteolytic activity and amino acid transport, but had no benefit to pod (sink) development. In general, the paper was well-written and results is quite significant, will be suitable published in the journal with the miner revision.

Minor comments:

  1. 应该更新文献综述。这篇论文几乎没有引用2018-2021年的最新论文(只有2019年和2020年的一篇)。作者在引言部分引用了该观点的最新论文。

回应)正如建议的那样,“引言部分”中的一些参考文献已被最近发表的论文所取代(参见修订稿中的参考文献 4-5、7-10、12-13、15、18、20 和 22-23) .

  1. 除了文献 25 和 27 之外,所有出版年份在参考列表中都被涂黑。参考 26 缺失。

回应)之前的手稿中存在某些编辑或格式错误。我们检查了参考文献列表,在修订​​稿中添加了参考文献 26。
